# A Collusion Attack on Stable Signature and a Defense using Domain-based Signature Assignment

## Abstract

Stable Signature is a recent watermarking framework based on latent diffusion models, which generates images with embedded signatures by fine-tuning the decoder. While prior work has shown that watermarks can be removed while maintaining visual quality by retraining the watermarked decoder with clean images, we demonstrate that collusion among multiple users poses a practical and severe threat. Our attack begins by averaging watermarked decoders, which already provides a strong initialization for watermark removal. With encoder access, this initialization can be further fine-tuned to significantly suppress the watermark signal. Even when the encoder is not available, colluders can expand their group size to achieve comparable effectiveness, highlighting the scalability of the attack. To defend against this threat, we propose a domain-based signature assignment mechanism. In this strategy, the watermarking service provider (e.g., one using Stable Signature) partitions the signature space into domains, requiring all users in the same domain to share a fixed set of domain-index bits in their signatures. Experiments show that the domain-index bits remain robust under the collusion attack when the encoder is not available. Our studies suggest that adopting the domain-based signature assignment and keeping the encoder confidential will be good practices when Stable Signature is used as a watermarking solution.

## 1 Introduction

Since the emergence of generative AI technology (Goodfellow et al., 2020; Ho et al., 2020; Saharia et al., 2022), it has been growing exponentially fast. The AI-generated content (AIGC) has already entered and made a deep impact on almost every aspect of human life, facilitated through the Internet. This rise in AIGC has also raised concerns about authenticity, attribution, and potential misuse. To address these concerns, watermarking techniques (Ramesh et al., 2022; Zhu et al., 2018; Tancik et al., 2020; Al-Haj, 2007; Zhao et al., 2023) have been proposed to embed signals that identify the origin of AI-generated outputs. Among these, Stable Signature (Fernandez et al., 2023) is a recent watermarking method designed for latent diffusion models (LDMs). It embeds a binary watermark into the decoder of a LDM, enabling per-user attribution. Its design avoids reliance on the encoder or inference-time post-processing, and it demonstrates strong robustness to image-level transformations such as cropping or compression. Because Stable Signature targets real-world deployment scenarios involving user-specific watermarking, evaluating its security under realistic threat models is essential.

Most existing watermark removal methods focus on image-level manipulations like cropping, compression, or adversarial editing (Zhao et al., 2024; Jiang et al., 2023; Saberi et al., 2023; Lukas et al., 2023). While these approaches highlight general vulnerabilities in image-level watermarking, they do not address model-level schemes like Stable Signature. The most relevant follow-up is the work by Hu et al. (2024), which shows that watermarks embedded by Stable Signature can be removed through fine-tuning. Their method works in both encoder-aware and encoder-agnostic settings, and demonstrates that retraining a watermarked decoder on clean data can effectively suppress the watermark signal. However, in the encoder-agnostic case, their approach requires significant computation: for each image, they solve an optimization problem to find a latent vector that reconstructs a given clean (non-watermarked) image. This makes the attack costly and difficult to scale, especially

when access to clean data is limited. Meanwhile, Stable Signature briefly discusses model collusion between two users, but provides neither in-depth analysis nor experimental studies. This leaves open the question of whether simple and scalable collusion attacks can also break watermarking in practical settings.

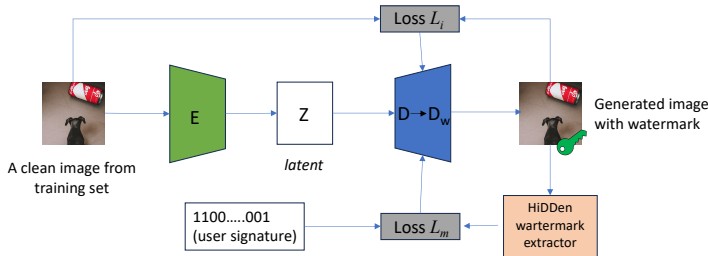

Figure 1: Overview of Stable Signature's Framework: fine-tuning the latent Decoder $D$ to the watermarked decoder $D_w$.

In this paper, we examine the gap in detail and present a systematic study of collusion in model-level watermarking. We assume that a group of users, each with their own watermarked decoder, collaborate to remove the watermark signal. Our attack begins by averaging their decoder weights, which already provides a strong initialization by partially canceling key-specific perturbations. We then propose two complementary methods to strengthen removal. First, when the encoder is available, this initialization can be fine-tuned to further suppress the watermark. Second, when the encoder is not accessible, colluders can enlarge their group size to achieve strong removal, demonstrating the scalability of collusion. To counter this threat, we propose a *domain-based signature assignment* approach. In this strategy, the watermarking service provider partitions the signature space into domains, requiring all users in the same domain to share a fixed set of domain-index bits. These shared bits induce correlated perturbations across decoders, which resist cancellation through averaging and thereby reduce the effectiveness of collusion without altering the core watermarking pipeline.

In summary, our contributions are as follows. First, we propose and systematically study, for the first time, a model collusion attack involving multiple users on Stable Signature framework, showing that model averaging followed by proper fine-tuning can effectively remove watermarks in both encoder-aware and encoder-agnostic settings. Second, we demonstrate that this collusion attack can be lightweight, as with an appropriately chosen loss function, fine-tuning requires only the colluders' watermarked images as training data, thereby avoiding the need for clean data or costly latent optimization. Third, we conduct a detailed empirical evaluation of Stable Signature framework under multi-user collusion, revealing that even a small number of colluding users can substantially weaken the watermark signal. Fourth, we propose a simple yet effective defense, Domain-based Signature Assignment, which introduces shared bits across watermark keys to prevent complete perturbation cancellation during averaging. Finally, we show that domain-based signature assignment improves robustness against collusion, particularly under encoder-agnostic conditions, while requiring only minimal modifications to the existing watermarking pipeline.

## 2 BACKGROUND AND RELATED WORK

Diffusion models (Sohl-Dickstein et al., 2015; Dhariwal & Nichol, 2021) generate high-quality images through iterative denoising in pixel space, but incur heavy computational cost. LDMs (Rombach et al., 2022) address this by operating in a compressed latent space, where a pre-trained autoencoder maps images into low-dimensional representations and a U-Net with cross-attention reverses a noise process to generate samples. The decoder reconstructs full-resolution outputs from latent representations, making it a natural point for embedding watermarks due to its direct influence on perceptual quality.

Stable Signature (Fernandez et al., 2023) fine-tunes the LDM decoder to embed a $k$-bit binary watermark using the HiDDen framework (Zhu et al., 2018). The training objective combines message loss $L_m$ with perceptual loss $L_i$ (Czolbe et al., 2020), controlled by $\lambda_i$, to balance robustness and

imperceptibility:

$$L = L_m + \lambda_i L_i. \tag{1}$$

The resulting decoder produces watermarked images that remain detectable under common transformations (cropping, compression, resizing). Detection relies on a watermark extractor that compares the extracted bits $m'$ to the original $m$, with thresholds such as 41/48 matching bits($\approx 85\%$) ensuring false positive rates below $10^{-6}$ (Hu et al., 2024).

Research on defeating watermarks falls into two categories. *Image-level attacks* manipulate the final output via transformations or adversarial removal (Zhao et al., 2024; Jiang et al., 2023; Saberi et al., 2023; Lukas et al., 2023). *Model-level attacks* alter the generative model itself, e.g., decoder purification or latent-based fine-tuning (Hu et al., 2024). Prior work on Stable Signature also briefly considered network-level threats such as model purification and pairwise collusion (Fernandez et al., 2023), but these experiments were limited in scope and did not explore larger-scale or systematic collusion. While effective, most existing approaches assume a single-user threat model, where each instance is attacked independently. This leaves a critical gap: coordinated adversaries with access to multiple uniquely watermarked decoders. Our work addresses this gap by studying collusion attacks, showing they systematically degrade watermark signals while preserving high image quality. Hu et al. (2024) demonstrated that optimization-based fine-tuning can be highly effective at watermark removal, but their method requires clean data and substantial computation, particularly in encoder-agnostic settings where optimization must be repeated per image. By comparison, collusion attack is lightweight, relying only on watermarked models and images from colluders, which makes it a practical and scalable threat that has been largely overlooked in prior work.

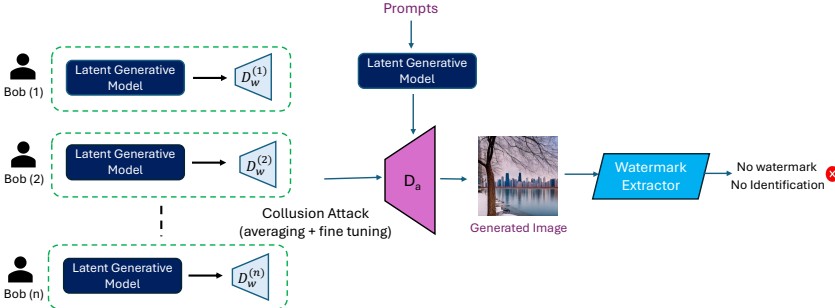

Figure 2: Overview of the collusion attack: Multiple users, each processing a uniquely watermarked decoder, collaborate to generate a new compromised decoder $D_a$. This colluded decoder $D_a$ is then used in the latent generative model pipeline to produce high-quality images that bypass the watermark extractor, resulting in failed detection and identification.

## 3 THREAT MODEL

We consider a scenario in which a total of $N$ users participate in the collusion as shown in Fig 2, each possessing a uniquely fine-tuned watermarked decoder $D_w^{(i)}$ that has been fine-tuned from a base decoder $D$ to embed a distinct 48-bit binary watermark key, where $i = 1, 2, \ldots, N$. The watermark is embedded directly into the decoder's parameters, enabling in-generation watermarking as proposed by Stable Signature. Note that our threat model follows a structure similar to that proposed in (Hu et al., 2024), which analyzes encoder-aware and encoder-agnostic attacks on watermarked diffusion decoders. However, instead of focusing on watermark removal via image supervision training, we propose a collusion attack leveraging multiple uniquely watermarked decoders.

### 3.1 ATTACKER'S GOAL

The goal of a colluding group of users is to remove the watermark from their decoders $D_w^{(i)}$. Specifically, the attackers aim to construct a new decoder $D_{\text{attack}}$ that can not only generate non-watermark images but also preserve the high perceptual quality of the generated images.

## 3.2 ATTACKER'S KNOWLEDGE

We assume the attacker has access to their own uniquely watermarked decoders $D_w^{(i)}$ for $i \in \{1, 2, \ldots, N\}$, a collection of independently generated watermarked images that serve as a training dataset, and the extractor model used to decode watermark bits from generated images.

Depending on the attacker's access to upstream components of the latent diffusion pipeline, we consider two scenarios. In the **encoder-agnostic** scenario, the attacker lacks access to the encoder and denoising modules but does have the latent vectors $z$ used for generation (e.g., captured during the forward pass of watermarked image generation). In the **encoder-aware** scenario the attacker has access to the encoder and diffusion process, enabling fast and scalable latent extraction for arbitrary images and allowing the attacker to use both watermarked images produced by $D_w^{(i)}$ and clean public datasets (e.g., ImageNet) for decoder fine-tuning, which facilitates more flexible and large-scale attacks.

## 3.3 ATTACKER'S CAPABILITY

We assume that the attackers are able to modify the parameters of their own decoder weights $D_w^{(i)}$.

In summary, this threat model reflects a realistic and potentially damaging scenario: a small number of colluding users, even with limited knowledge of the full pipeline, may still successfully remove the watermark while preserving image quality.

## 4 COLLUSION ATTACK METHOD

Based on the threat model introduced in Section III, we now present our collusion watermark removal strategy. In this setting, each colluding user holds a uniquely watermarked decoder $D_w^{(i)}$, whose weights we denote as $W^{(i)} = W + \Delta^{(i)}$, where $W$ represents the parameters of the clean (non-watermarked) decoder, and $\Delta^{(i)}$ is the perturbation introduced to embed the watermark key. The group of attackers aims to construct a new decoder that suppresses watermark signals while maintaining image quality.

Let $W \in \mathbb{R}^d$ denote the parameter vector of the clean decoder. The watermarked decoder for the $i$-th user is given by:

$$W^{(i)} = W + \Delta^{(i)}, \tag{2}$$

where $\Delta^{(i)} \in \mathbb{R}^d$ is the watermark-specific perturbation, and $d$ is the total number of decoder parameters. The addition is performed element-wise.

Since the 48-bit watermark keys are randomly sampled per user and the fine-tuning process preserves generative quality, the perturbations $\Delta^{(i)}$ are expected to be small in magnitude and symmetrically distributed:

$$\Delta^{(i)} \in [-\alpha, \beta]^d, \quad \text{with } \alpha \approx \beta > 0 \text{ and relatively minor in scale.} \tag{3}$$

In practice, the outputs of different $D_w^{(i)}$ appear perceptually similar, as shown in Fig. 3, reflecting that $\Delta^{(i)}$ does not significantly shift the model's image distribution.

Now consider a scenario where $N$ users collude by averaging their watermarked model weights. The resulting decoder has weights:

$$W_{\text{avg}} = \frac{1}{N} \sum_{i=1}^{N} W^{(i)} = W + \frac{1}{N} \sum_{i=1}^{N} \Delta^{(i)}. \tag{4}$$

If the perturbations $\Delta^{(i)}$ are approximately zero-mean and uncorrelated across users, the aggregated perturbation term tends to cancel out as $N$ increases:

$$W_{\text{avg}} \approx W. \tag{5}$$

This implies that averaging multiple watermarked decoders can reduce or eliminate the embedded watermark signal, since user-specific perturbations destructively interfere under ideal statistical assumptions. As shown in Fig. 4a, collusion already lowers the bit accuracy to 0.81, much lower than

other attack baselines. This substantial drop, however, still leaves a detectable watermark, which motivates us to go beyond averaging.

To address the limitations of averaging alone, we propose a two-stage attack method under both encoder-aware and encoder-agnostic settings. In the first stage (Averaging), we compute $W_{\text{avg}}$ as the average of the weights from $N$ watermarked decoders, providing an initialization that weakens the watermark signal by partially canceling diverse perturbations. In the second stage (Fine-tuning), the decoder initialized with $W_{\text{avg}}$ is further optimized using a dataset of watermarked images generated by $D_w^{(i)}$ (detailed in the next subsection), which helps remove residual watermark signals while preserving image quality. This strategy leverages the fact that averaging alone can suppress much of the watermark signal but not always completely eliminate it, whereas fine-tuning completes the removal by explicitly optimizing the model in the image space.

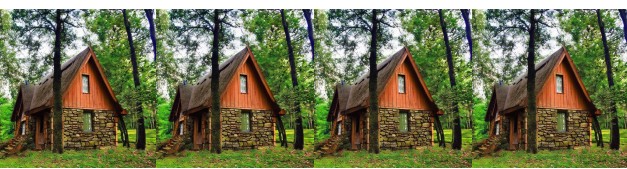

Figure 3: Images generated in order (from left to right) by Decoder with $W$, $D_w^{(1)}$, $D_w^{(2)}$, Decoder with $W_{\text{avg}}$.

## 4.1 ATTACKER DATASET SETUP

To fine-tune the decoder after getting $D_{\text{avg}}$ according to equation equation 4, we next construct a dataset by $D_w^{(i)}$ following Stable Signature's pipeline. This approach allows attackers to generate training data without relying on original or external datasets.

Each data in the dataset consists of a latent vector $z_j$ and its corresponding watermarked image $x_{w,j}$, such that

$$x_{w,j}^{(i)} = D_w^{(i)}(z_j), \quad \forall i \in \{1, \ldots, N\}, \forall j \in \{1, \ldots, M_i\}. \tag{6}$$

As shown in equation equation 6, $(z_j, x_{w,j}^{(i)})$ denotes the $j$-th latent–image pair generated using $D_w^{(i)}$, where $j \in \{1, \ldots, M_i\}$ and $M_i$ is the number of samples generated by user $i$. These latent vectors $z_j$ are captured using diverse prompts and noise seeds to ensure broad coverage of the latent space.

As shown in Figure 4b, latent vectors $z_j$ are generated and passed through different watermarked decoders $D_w^{(i)}$ to produce the corresponding images $x_{w,j}^i$. The resulting $(z_j, x_{w,j}^{(i)})$ pairs are collected and used to supervise decoder fine-tuning.

Note that the $(z_j, x_{w,j}^{(i)})$ pairs are particularly for the encoder-agnostic setting, since there is no encoder for latent mapping. For the encoder-aware setting, we use only watermarked images $x_w^{(i)}$ as our dataset because the attackers have latent mapping ability with the encoder.

## 4.2 POST-COLLUSION FINE-TUNING

The final step of our attack refines the averaged decoder $D_{\text{avg}}$ by fine-tuning the attacker's dataset described in the previous subsection. The goal is to further remove residual watermark signals while preserving image quality.

We denote the fine-tuned decoder as $D_a$. The fine-tuning process optimizes $D_a$ using gradient descent on a reconstruction loss between the decoder's output and the target watermarked image. Specifically, for each latent–image pair $(z_j, x_{w,j}^{(i)})$, the decoder is trained to minimize the following objective:

$$\mathcal{L}_{\text{total}} = \mathcal{L}_{\text{rec}} + \lambda \cdot \mathcal{L}_{\text{perc}}, \tag{7}$$

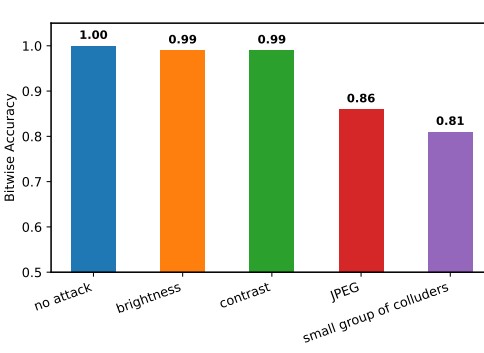

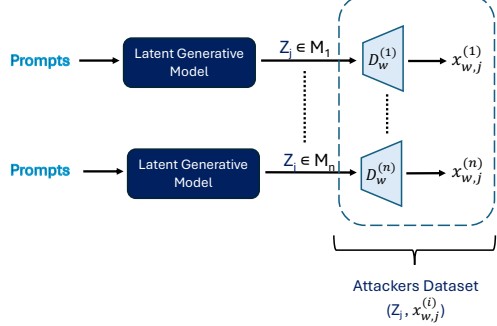

(a) Bitwise accuracy across different attacks.

(b) Generation of attacker data: same prompts through different decoders.

Figure 4: Evaluation of robustness and attacker data setup.

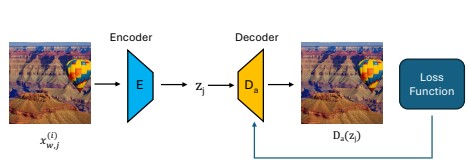

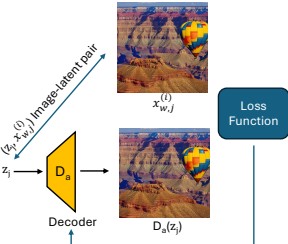

(a) Overview of the encoder-aware scenario: Encoder $E$ is available to the attacker, each image $x_{w,j}^{(i)}$ in the attacker's dataset passes through the encoder to provide $z_j$. This latent is passed through the attack decoder. The weights of the attack decoder are updated based on the loss function.

(b) Overview of the encoder-agnostic scenario: Encoder $E$ is not available to the attacker, the saved latent-image pair $(z_j, x_{w,j}^{(i)})$. This latent is passed through the attack decoder. The weights of the attack decoder are updated based on the loss function.

Figure 5: Illustration of encoder-aware and encoder-agnostic attack scenarios.

where $\mathcal{L}_{\text{rec}}$ is a pixel-wise loss (e.g., MSE), $\mathcal{L}_{\text{perc}}$ is a perceptual loss, and $\lambda$ controls the weight of perceptual alignment. We use perceptual loss by default, but our framework supports other choices such as SSIM (Wang et al., 2004), LPIPS (Zhang et al., 2018), or a combination of these.

This fine-tuning is performed under two scenarios. In the encoder-aware case (see Fig. 5a), the attacker has access to the encoder of the latent diffusion model and therefore retains only the watermarked images $x_w^{(i)}$, extracting their latent vectors on-the-fly via the encoder; this ensures the latents are consistent with the model's encoding process and allows the attack decoder $D_a$ to be trained on freshly encoded pairs $(E(x_w^{(i)}), x_w^{(i)})$ (see Algorithm 1). In the encoder-agnostic case (see Fig. 5b), the attacker lacks encoder access and instead uses precomputed latent–image pairs $(z_j, x_{w,j}^{(i)})$ saved during dataset construction; this avoids re-encoding cost and is more efficient for resource-constrained adversaries (see Algorithm 2).

We observe that watermark removal in the encoder-agnostic scenario is less effective compared to the encoder-aware case. However, attackers can amplify the effect by enlarging the colluder group. With more colluders, the residual watermark is further suppressed (Table 1). To counter such a strong attack, we introduce a defense strategy, described in the following section.

## 5 DEFENSE VIA DOMAIN INDEX ASSIGNMENT

To mitigate strong collusion attacks, we propose a key assignment strategy called *Domain Index Assignment*. This method introduces correlation across watermark keys by sharing partial key bits among users, thereby breaking the assumption that $\Delta^{(i)}$ are independent and zero-mean. This assumption is critical to the success of averaging-based attacks. This approach is similar to the idea of

**Algorithm 1** Encoder-aware collusion fine-tuning

**Require:** Images $x_w^{(i)}$ from attackers' dataset, Decoder weights $W^{(1)}, W^{(2)}, \ldots, W^{(n)}$
1: $W_{\text{avg}} \leftarrow \frac{1}{n} \sum_{i=1}^{n} W^{(i)}$
2: $W_a \leftarrow W_{\text{avg}}$
3: **repeat**
4:     Compute loss $\mathcal{L}$ (Eq. equation 7)
5:     $W_a \leftarrow W_a - \eta \nabla_{W_a} \mathcal{L}$
6: **until** convergence
7: Save final decoder $D_a$ with weights $W_a$

**Algorithm 2** Encoder-agnostic collusion fine-tuning

**Require:** Latent-Image pairs $(E(x_w^{(i)}), x_w^{(i)})$, Decoder weights $W^{(1)}, W^{(2)}, \ldots, W^{(n)}$
1: $W_{\text{avg}} \leftarrow \frac{1}{n} \sum_{i=1}^{n} W^{(i)}$
2: $W_a \leftarrow W_{\text{avg}}$
3: **repeat**
4:     Compute loss $\mathcal{L}$ (Eq. equation 7)
5:     $W_a \leftarrow W_a - \eta \nabla_{W_a} \mathcal{L}$
6: **until** convergence
7: Save final decoder $D_a$ with weights $W_a$

IP address allocation within a network: the high-order bits (network prefix) are fixed to define the subnet, while the low-order bits (host ID) are varied per user. In our context, domain-based signature assignment ensures that part of the watermark key is consistent across users. This bit-sharing prevents cancellation through averaging.

## 5.1 DOMAIN-BASED SIGNATURE ASSIGNMENT

Each user-specific decoder $D_w^{(i)}$ is embedded with a 48-bit key $k^{(i)} \in \{0,1\}^{48}$. Instead of sampling all 48 bits independently, the key is split into

$$k^{(i)} = \left[k_{\text{fixed}}, \ k_{\text{rand}}^{(i)}\right], \tag{8}$$

where $k_{\text{fixed}} \in \{0,1\}^n$ is a secret prefix shared by all users and $k_{\text{rand}}^{(i)} \in \{0,1\}^{48-n}$ is user-specific. This reduces the key space to $2^{48-n}$, balancing robustness and uniqueness: larger $n$ yields stronger correlation (robustness), smaller $n$ more entropy.

## 5.2 IMPACT ON AVERAGING ATTACKS

Without defense, keys are sampled uniformly, giving perturbations $\Delta^{(i)} \sim \mathcal{U}(-\alpha, \alpha)^d$, so that $\frac{1}{N} \sum_{i=1}^{N} \Delta^{(i)} \approx 0$ and averaging cancels the watermark. With domain-based assignment, perturbations split as

$$\Delta^{(i)} = \Delta_{\text{shared}} + \Delta_{\text{unique}}^{(i)}, \tag{9}$$

where $\Delta_{\text{shared}}$ is fixed across users. The average becomes

$$\frac{1}{N} \sum_{i=1}^{N} \Delta^{(i)} = \Delta_{\text{shared}} + \frac{1}{N} \sum_{i=1}^{N} \Delta_{\text{unique}}^{(i)}, \tag{10}$$

so the first term preserves watermark bias even as the second term vanishes. Thus averaging no longer cancels the signal, invalidating the core assumption behind collusion attacks.

## 6 EXPERIMENTAL RESULTS

We conduct a comprehensive set of experiments to evaluate both the proposed collusion attacks and our defense. Our study covers encoder-aware and encoder-agnostic scenarios, small and large groups of colluders, and comparisons against standard image-level attacks (cropping, JPEG compression, brightness adjustment). In addition, we examine the effectiveness of domain-based signature assignment under different attack settings.

To keep the main paper focused, we report the most representative results here and include additional experiments: the impact of larger colluder-generated datasets (Fig 7) and varying numbers of fixed bits in the defense (Fig 8) in the Appendix. Together, these results provide a complete picture of the threat and demonstrate the robustness of our proposed defense.

## 6.1 EXPERIMENTAL SETUP

**Dataset**: We used 3000 image–latent pairs generated from 1000 prompts, each rendered with three different decoders to simulate randomly watermarked outputs. Source images were 512×512; for encoder-aware training they were downsampled to 256×256, while encoder-agnostic training used the original 512×512 images with 64×64 latents. Watermarked decoders were created by fine-tuning the clean Stable Diffusion-2 decoder on 2000 MS-COCO images, yielding 10 decoders each with a unique 48-bit watermark. A disjoint test set of 100 images per method was generated from non-overlapping prompts.

**Parameter Settings**: Training and evaluation were performed on an NVIDIA RTX 4070 Ti GPU. We followed Stable Signature hyperparameters (architecture, learning rate, batch size, optimizer) and employed MSE as reconstruction loss with Watson-VGG as perceptual loss. In our experiments, we define a small colluder group as 3 users and a large colluder group as 10 users.

**Baselines**: We compared against three per-image removal techniques such as contrast, JPEG compression, and brightness adjustment.

**Metrics**: Performance was measured by bitwise accuracy (fraction of extracted bits matching the ground-truth key) and image quality using FID, SSIM, and PSNR. For each test image, outputs were compared to the corresponding watermarked Stable Diffusion-2 image generated with the same seed (Hu et al., 2024). Final results report averages over the 100 test images.

Table 1: Comparison of attack performance across different settings. ($\downarrow$ indicates lower is better; $\uparrow$ indicates higher is better.)

| Method | FID $\downarrow$ | PSNR $\uparrow$ | SSIM $\uparrow$ | Bit Acc $\downarrow$ |
|---|---|---|---|---|
| $W_{avg}$ of large group of colluders | 6.27 | 30.57 | 0.90 | 0.65 |
| $W_{avg}$ of small group of colluders | 10.95 | 30.24 | 0.90 | 0.75 |
| Encoder-aware under small group of colluders | 19.30 | 26.61 | 0.79 | 0.61 |
| Encoder-agnostic under small group of colluders | 14.68 | 27.97 | 0.85 | 0.74 |

## 6.2 COLLUSION EFFECTIVENESS

Table 2: Comparison of attack performance with domain-based index assignment

| Method | FID $\downarrow$ | PSNR $\uparrow$ | SSIM $\uparrow$ | Bit Acc $\downarrow$ | Last 10-Bit Acc $\downarrow$ |
|---|---|---|---|---|---|
| $W_{avg}$ of large group of colluders | 4.46 | 31.70 | 0.92 | 0.92 | 0.64 |
| Encoder-aware | 17.92 | 26.84 | 0.79 | 0.64 | 0.67 |
| Encoder-agnostic | 20.49 | 28.36 | 0.89 | 0.93 | 0.68 |

We first evaluate collusion attacks under small group of colluders. As shown in Table 1, both encoder-aware and encoder-agnostic variants significantly weaken the watermark while preserving high image quality, and Fig. 6a shows that the generated images remain visually close to Stable Diffusion 2 outputs. With encoder access, fine-tuning poses a severe threat, driving bit accuracy down from 1.00 to 0.61 with SSIM 0.79 and FID 19.30. The encoder-agnostic case achieves smaller gains from fine-tuning, yet its danger should not be underestimated: by simply expanding the colluder group, accuracy drops even further to 0.65, exposing a critical vulnerability in watermark robustness.

Table 3: Comparison with other attacks

| Attack Method | FID $\downarrow$ | PSNR $\uparrow$ | SSIM $\uparrow$ | Bit accu $\downarrow$ |
|---|---|---|---|---|
| Brightness | 185.45 | 5.07 | 0.36 | 0.99 |
| Contrast | 42.10 | 16.61 | 0.64 | 0.99 |
| JPEG compression | 29.10 | 28.56 | 0.83 | 0.86 |
| $W_{avg}$ of small group of colluders | 10.95 | 30.24 | **0.9** | 0.75 |
| $W_{avg}$ of large group of colluders | **6.27** | **30.57** | **0.9** | 0.65 |
| Model purification (MP) | 18.66 | 26.99 | 0.80 | 0.89 |
| Encoder-aware under small group of colluders | 19.30 | 26.61 | 0.79 | **0.61** |
| Encoder-agnostic under small group of colluders | 14.68 | 27.97 | 0.85 | 0.74 |

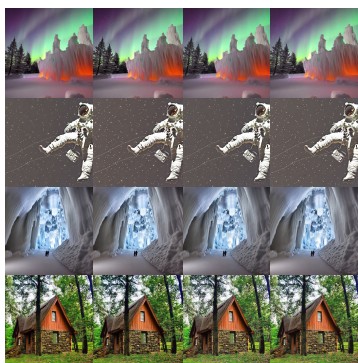 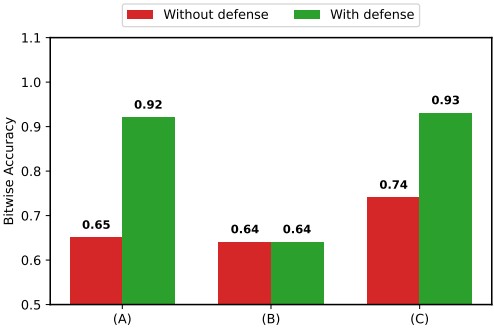

(a) Comparison of different attacks: images generated in the order (left to right) by Stable Diffusion 2 model, Stable Signature watermarked decoder $(D_w^{(1)})$, our encoder-aware attack decoder, and our encoder-agnostic attack decoder.

(b) Performance of domain-based signature assignment: (A) is the comparison between $W_{\text{avg}}$ of large group of colluders , (B) is the comparison under encoder-aware, and (C) is the comparison under encoder-agnostic.

Figure 6: Visual and quantitative evaluation of collusion attacks and our defense.

## 6.3 DOMAIN-BASED ASSIGNMENT DEFENSE EFFECTIVENESS

To evaluate our defense, we tested watermarked models under collusion using domain-based signature assignment with 38 fixed bits and 10 random bits (Table 2, Fig. 6b). The results show a clear improvement: bit accuracy remains as high as 0.93 in the encoder-agnostic case and 0.92 for the large group of colluders' averaged decoder $D_{\text{avg}}$ with fixed bits, compared to only 0.65 without defense. Even when attackers target the last 10 bits, the full 48-bit key persists, demonstrating strong protection against collusion while preserving image quality.

Our defense is especially effective against large-scale collusion in the encoder-agnostic scenario, where attackers are limited to latent–image pairs. In contrast, the encoder-aware setting is less robust, since access to the encoder enables attackers to generate diverse training pairs that strengthen fine-tuning by using clean images. This gap highlights an important takeaway for model providers: restricting access to the encoder is critical for maintaining watermark robustness.

Compared to prior work (Table 3), our method consistently resists collusion attacks while sustaining high perceptual quality. By embedding shared bits across users, domain-based signature assignment blocks the perturbation cancellation that underlies averaging attacks.

## 6.4 REPRODUCIBILITY

We are committed to reproducibility. Upon acceptance, we will release the full codebase, pretrained models, and data generation scripts, together with instructions for reproducing all reported results, including the collusion attacks, the proposed defense, and the evaluation metrics (bitwise accuracy, FID, PSNR, SSIM).

## 7 CONCLUSION

We presented a systematic study of collusion attacks on Stable Signature. By averaging watermarked decoders and applying fine-tuning, colluding users can effectively weaken or remove embedded watermarks, even without access to clean data. We further showed that enlarging the colluder group strengthens the attack under encoder-agnostic settings. To counter this threat, we introduced domain-based signature assignment, which preserves robustness against collusion by embedding shared bits across users. Our findings highlight both the practicality of collusion attacks and the importance of restricting encoder access and adopting domain-based defenses for secure watermarking deployment.

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

# A  APPENDIX

## A.1  ADDITIONAL EXPERIMENTAL RESULTS

### A.1.1  EFFECT OF LARGER COLLUDER-GENERATED DATASETS

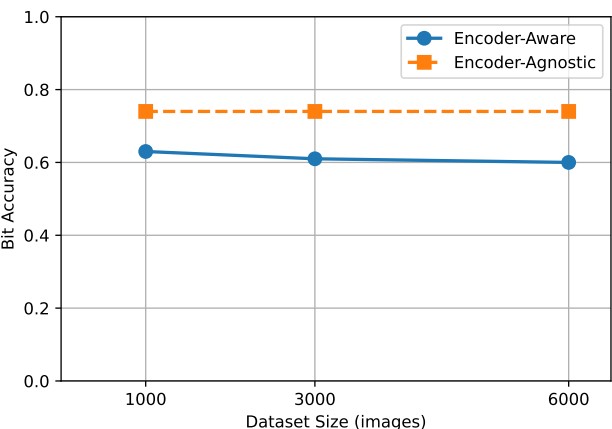

Figure 7: Bitwise accuracy of encoder-aware and encoder-agnostic collusion attacks under varying dataset sizes (1000, 3000, and 6000 images). Results show that enlarging the dataset has little effect on performance, confirming that expanding the number of colluders is more effective than scaling individual datasets.

We further investigate whether increasing the dataset size of the colluding users improves watermark removal. Figure 7 shows bitwise accuracy across datasets of different sizes (1000, 3000, and 6000 images) under both encoder-aware and encoder-agnostic settings. The results indicate that extending the colluders' dataset size does not significantly improve performance: the curves remain nearly flat across different sizes.

This observation suggests that dataset scaling provides limited benefit to colluders, and that the key factor for effectiveness lies instead in increasing the number of colluders.

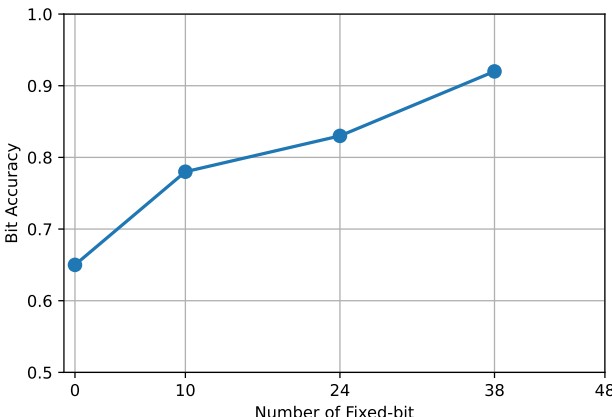

Figure 8: Bit accuracy under collusion attack with different numbers of fixed bits. More fixed bits result in stronger robustness against large group collusion averaging.

### A.1.2 DEFENSE UNDER DIFFERENT DOMAIN ASSIGNMENTS

To further study the robustness of domain-based signature assignment, we vary the number of fixed bits from 0 (no defense) to 38 (strongest setting used in the main paper). Figure 8 shows the bit accuracy under collusion attacks as the number of fixed bits increases. The trend reveals that even a small number of fixed bits (e.g., 10) already improves robustness substantially compared to large group collusion, and the protection continues to strengthen as more bits are fixed. This supports our intuition: increasing correlation across users' signatures preserves a stronger shared watermark signal that resists cancellation during averaging.

## B USE OF LARGE LANGUAGE MODELS (LLMS)

In accordance with ICLR 2026's policy on LLM usage, we disclose that we used a large language model (OpenAI's ChatGPT) solely for improving grammar, phrasing, and readability of the manuscript. All research ideas, experimental design, implementations, results, analyses, and mathematical formulations are original work by the authors. The LLM was not used to generate research content, code, or interpretations of results. The authors take full responsibility for the accuracy and integrity of the paper's content.

