# OpenReview forum: "A COLLUSION ATTACK ON STABLE SIGNATURE AND A DEFENSE USING DOMAIN-BASED SIGNATURE ASSIGNMENT"
_ICLR.cc/2026/Conference — ICLR 2026 Conference Withdrawn Submission_

### Official Review · Reviewer_Ya4P · 2025-10-20

**Soundness:** 3
**Presentation:** 3
**Contribution:** 2
**Rating:** 4
**Confidence:** 4

**Summary:**

The paper proposes an approach to attack stable signature based watermarking system by proposing an approach where a small number of users could collude to update the deocder. The authors also propose a defense for this attack. They show that this attack works well in practice when we assume access to the decoder. The defense as well shows promising results on everything except on the encoder aware setting.

**Strengths:**

- The attack is simple and intuitive.
- The paper is also well written and easy to follow.
- I appreciate that the paper introduces and attack as well as a technique to avert the threat posed by the attack.
- The attack seems to be practical as with only 3 colluders they are able to achieve significant performance drop of the watermarking system.

**Weaknesses:**

- The major weakness is that the authors assume that the attack will have access to the decoder weights as well as the z vector during generation. This in my opinion doesn’t represent a real world setting wherein the model owner would control the entire generation pipeline. A stronger attack would assume knowledge of the decoder architecture and or use a proxy decoder from another model to remove the watermark post-hoc.
- The authors also give a lot of emphasis on encoder aware or agnostic but for me this distinction is not as important since the authors are already making white box access assumption on the decoder and latent variable, thus assuming access to the encoder is not big. This is especially important since the proposed defense does not work well in the encoder aware attack.
- PSNR values below 30 seem to be overly large especially for the encoder aware setting.
- Baseline comparisons are lacking. The authors have cited multiple papers on watermark removal that exist but have not compared with them.
- It would be nice to see how Bit Acc translated into attack success rates based on the thresholds.

**Questions:**

- How does the attack generalize beyond the users that were used for attacking. For example if I average the weight for 3 users, will I be able to use the final decoder to attack a 4th unseen user?
- How does this attack generalize beyond Stable Signature or is it only applicable for stable signature? Can it generalize to other decoder specific watermarking systems?

---

### Official Review · Reviewer_7KKq · 2025-10-22

**Soundness:** 2
**Presentation:** 1
**Contribution:** 1
**Rating:** 2
**Confidence:** 4

**Summary:**

This paper investigates a new and realistic threat to stable signature, a watermarking framework designed for latent diffusion models. It introduces the model-level collusion attack. Colluders can collaborate to remove watermarks by combining their models and optionally fine-tuning the result. To defend against thisl, the paper proposes a domain-based signature assignment mechanism that makes watermark keys partially shared across users, preventing effective calcellation during averaging.

**Strengths:**

1. The paper systematically study model-level collusion attacks on stable signature.
2. The proposed collusion attack is conceptually simple yet effective.

**Weaknesses:**

1. The entire study focuses only on stable signature. Results may not extend to newer or structurally different watermarking schemes.
2. The threat model depends on assumptions about user access and linear watermark encoding.
3. Limited ablation depth and missing baselines.

**Questions:**

1. How do results scale when colluder models come from different training checkpoints or noise schedules?

---

### Official Review · Reviewer_WgTs · 2025-10-30

**Soundness:** 2
**Presentation:** 3
**Contribution:** 2
**Rating:** 2
**Confidence:** 5

**Summary:**

This paper introduces a two-stage collusion attack to remove the watermark signal from the watermarked decoder produced by Stable Signature. The proposed method first averages the weights across multiple watermarked decoders and then finetunes the averaged decoder to further remove the watermark signal. The author also purposes a defense using domain-based signature assignment to mitigate this attack. The experimental results show that the proposed method outperforms multiple baseline methods and can effectively remove watermark from Stable Signature.

**Strengths:**

1. The proposed defense, Domain-based Signature Assignment, is straightforward to implement and conceptually simple. It effectively targets the core assumption of the averaging-based attack by introducing correlations.
2. The paper is well-written and structured. The attack methodology and defense mechanism are explained clearly, and the figures effectively illustrate the core concepts.
3. The authors evaluate their attack under both encoder-aware and encoder-agnostic settings, analyze the effect of the number of colluders, and compare it against basic image manipulation baselines.

**Weaknesses:**

1. The central premise of the threat model is questionable. It assumes a service provider issues a unique watermark to each user. In practice, a provider aiming to identify their own generated content would likely use a single, secret watermark for all users. This would be more robust and would render the proposed collusion attack impossible, as there would be no different watermarks to average out. The motivation for per-user watermarking in a way that enables this attack is not well-justified.
2. The effectiveness of the model averaging relies on the assumption in Equation 3 that the watermark perturbations $\Delta^{(i)}$ are small, symmetrically distributed, and uncorrelated, causing them to cancel out. This is presented as an intuition without theoretical proof or strong empirical validation. It is not clear if the fine-tuning process of Stable Signature necessarily produces such well-behaved perturbations.
3. The paper compares the collusion attack only against very basic image-level attacks (brightness, contrast, JPEG) and briefly mentions model purification. The field of watermark removal is much broader, with more advanced and relevant attacks available. For instance, methods like CtrlRegen and other optimization-based removal techniques should have been included to provide a more meaningful comparison of the attack's potency and the defense's robustness.
4. The work is focused exclusively on the Stable Signature framework. While interesting, this makes the contribution very narrow.

**Questions:**

1. Could you elaborate on the practical scenario that motivates the threat model? Why would a service provider choose to deploy unique, user-specific watermarks in a manner that exposes them to this collusion attack, rather than using a single provider-level watermark? And are there any real-world applications of this threat model?
2. Can you provide theoretical or empirical evidence to support the assumption that the watermark perturbations $\Delta^{(i)}$ are approximately zero-mean and uncorrelated across users, as stated in Section 4? How sensitive is the attack's success to this assumption?
3. To better situate the paper's contribution, would you consider comparing your attack and defense against stronger, more recent watermark removal baselines (e.g., CtrlRegen)?
4. How could the proposed collusion attack and the domain-based defense be adapted or generalized to other watermarking methods that modify model weights? Does the core idea hold for frameworks other than Stable Signature?

---

### Official Review · Reviewer_Z4NL · 2025-10-31

**Soundness:** 1
**Presentation:** 3
**Contribution:** 1
**Rating:** 2
**Confidence:** 4

**Summary:**

The paper proposes a collusion attack scenario in which multiple users have access to watermarked images (or watermarked models). It introduces a two-stage strategy for removing the Stable Signature watermark: (a) Averaging the decoder parameters across users to initialize the model, and (b) Fine-tuning the averaged model using the collected watermarked images. To defend against this attack, the paper further proposes a domain-based signature assignment mechanism. In this approach, certain reserved server bits are made consistent across all users, thereby mitigating the effectiveness of the averaging-based attack.

**Strengths:**

1. The writing is clear and easy to follow, and the overall structure of the paper is well-organized.

2. The proposed collusion attack is novel. Previous defense methods may fail when multiple users collude to perform an attack, making this an important and valuable research direction for the community.

**Weaknesses:**

1. The scenario considered in this paper is quite restricted. For the encoder-agnostic case, it is not realistic, see Question 1. For the encoder-aware case, watermark providers can simply protect their systems by black-boxing the encoder and decoder, making the proposed attack inaccessible. This limitation significantly constrains the practical impact of the method.

2. The paper does not sufficiently engage with recent studies on watermark removal, including but not limited to [1–4].

3. The experiments in this paper are not convincing:

    a.  The authors use only 100 images for testing, which makes the results statistically unreliable. Metrics such as FID are highly sensitive to the number and diversity of evaluation images.

    b.  The comparison with other attack methods is limited. The paper only considers three traditional attacks and model purification, while many other relevant attacks exist, such as Rotation, Crop, Erase, Blur, Gaussian Noise, Diffusion Purification [1], Diffusion Regeneration [2], Rinsing Regeneration [3], and Averaging Attack [4].

4. The defense strategy of fixing an n-bit key comes at the cost of reducing the robustness of the watermark itself. The number of valid bits for detection decreases to 48 – n. Although this design improves robustness to the specific attack proposed in the paper, it should make the watermark less robust against other types of attacks.

[1] Weili Nie, Brandon Guo, Yujia Huang, Chaowei Xiao, Arash Vahdat, and Anima Anandkumar. Diffusion models for adversarial purification. In International Conference on Machine Learning (ICML), 2022.

[2] Xuandong Zhao, Kexun Zhang, Yu-Xiang Wang, and Lei Li. Generative autoencoders as watermark attackers: Analyses of vulnerabilities and threats. 2023

[3] Bang An, Mucong Ding, Tahseen Rabbani, Aakriti Agrawal, Yuancheng Xu, Chenghao Deng, Sicheng Zhu, Abdirisak Mohamed, Yuxin Wen, Tom Goldstein, and Furong Huang. Benchmarking the robustness of image watermarks, 2024.

[4] Pei Yang, Hai Ci, Yiren Song, and Mike Zheng Shou. Can simple averaging defeat modern watermarks? Advances in Neural Information Processing Systems, 37:56644–56673, 2024.

**Questions:**

1. I find the encoder-agnostic scenario unrealistic. It is unclear under what circumstances one would have access to the latent vectors and decoder weights but not the encoder. In other words, why would the watermark provider share both the latent representations and the decoder with an external party who could potentially launch an attack? If the provider intends to protect the model, they could simply make it black-box, returning only the watermarked images.
2. Are Table 1 and a portion of Table 3 identical? If so, it would be better to remove Table 1 to avoid redundancy.

---

### Note · Authors · 2025-11-13

I have read and agree with the venue's withdrawal policy on behalf of myself and my co-authors.